# Cryo-EM structures of the channelrhodopsin ChRmine in lipid nanodiscs

Kyle Tucker[1,2,3], Savitha Sridharan[1,2], Hillel Adesnik [1,2] ✉ & Stephen G. Brohawn [1,2,3] ✉

Microbial channelrhodopsins are light-gated ion channels widely used for optogenetic manipulation of neuronal activity. ChRmine is a bacteriorhodopsin-like cation channelrhodopsin (BCCR) more closely related to ion pump rhodopsins than other channelrhodopsins. ChRmine displays unique properties favorable for optogenetics including high light sensitivity, a broad, red-shifted activation spectrum, cation selectivity, and large photo-currents, while its slow closing kinetics impedes some applications. The structural basis for ChRmine function, or that of any other BCCR, is unknown. Here, we present cryo-EM structures of ChRmine in lipid nanodiscs in apo (opsin) and retinal-bound (rhodopsin) forms. The structures reveal an unprecedented trimeric architecture with a lipid filled central pore. Large electronegative cavities on either side of the membrane facilitate high conductance and selectivity for cations over protons. The retinal binding pocket structure suggests channel properties could be tuned with mutations and we identify ChRmine variants with ten-fold decreased and two-fold increased closing rates. A T119A mutant shows favorable properties relative to wild-type and previously reported ChRmine variants for optogenetics. These results provide insight into structural features that generate an ultra-potent microbial opsin and provide a platform for rational engineering of channelrhodopsins with improved properties that could expand the scale, depth, and precision of optogenetic experiments.

Rhodopsins are photoreceptor proteins divided into type I (microbial) and type II (animal) groups[1–3]. Rhodopsins consist of an opsin protein with seven transmembrane helices (TM1-TM7) rendered photosensitive by a covalently bound retinal chromophore[2–6]. Light-induced retinal isomerization results in protein conformational changes coupled to downstream functions. Animal type II rhodopsins are G-protein coupled receptors in which isomerization of 11-cis-retinal to all-trans-retinal initiates signaling through heterotrimeric G-proteins and arrestin. Microbial type I rhodopsins are more diverse; isomerization of all-trans retinal to 13-cis-retinal results in protein conformational changes that are coupled to enzymatic, ion pump, or ion channel activities in different proteins[2–6].

Ion pump and ion channel rhodopsins are widely utilized in optogenetics where their heterologous expression in targeted cells enables control of membrane potential and electrical excitability with light[3,7]. Targeted neuronal activation is typically achieved with depolarizing cation-selective channelrhodopsins, while silencing is achieved with hyperpolarizing pumps or anion-selective channelrhodopsins[3]. Two decades since the initial characterization of a cation-selective channelrhodopsin (CCR) from the chlorophyte algae *Chlamydomonas*

---

[1]Department of Molecular & Cell Biology, University of California Berkeley, Berkeley, CA 94720, USA. [2]Helen Wills Neuroscience Institute, University of California Berkeley, Berkeley, CA 94720, USA. [3]California Institute for Quantitative Biology (QB3), University of California, Berkeley, CA 94720, USA. ✉e-mail: hadesnik@berkeley.edu; brohawn@berkeley.edu

*reinhardtii* ChR1[8], the optogenetic toolbox has been greatly expanded by efforts to discover or engineer novel channelrhodopsins with varied properties[3]. Key features distinguishing channelrhodopsins include selectivity for different ions, kinetics of channel opening and closing, light sensitivity, spectral properties, and photocurrent magnitude. Most of the currently utilized depolarizing channelrhodopsins are derived from genes from chlorophyte algae (including ChR2/CHETA[9,10], CoChR[11], Chronos/ChroME[11–13], and Chrimson[11]). However, a distinct family of channelrhodopsins was recently identified in cryptophyte algae and termed bacteriorhodopsin-like cation channelrhodopsins (BCCRs). BCCRs more closely resemble archaeal pump rhodopsins, such as *Halobacterium salinarum* bacteriorhodopsin, than CCRs in sequence and mechanistic properties[14–16]. Gating of the BCCR GtCCR2 involves proton transfer between the retinal Schiff base and residues in a manner analogous to how proton transport is achieved in bacteriorhodopsin[16]. Still, the mechanistic basis for BCCR channel activity in the context of higher homology to pump rhodopsins is not fully understood.

ChRmine is a BCCR identified through structure-based sequence database mining that is distinct among optogenetic tools for its combination of high light sensitivity, cation over proton selectivity, large photocurrents, and broad, red-shifted action spectrum[17]. These properties have enabled previously difficult or intractable experiments including holographic stimulation of large numbers of cortical neurons and transcranial stimulation of deep brain circuits[17–19]. Still, some properties of ChRmine limit its utility in certain applications, including a slow closing rate that precludes accurate command of high frequency spike trains and a broad absorbance spectrum that limits spectral multiplexing with other tools[13,17]. Structure-based engineering of other channelrhodopsins has successfully tuned channel properties[12,13,20–23], but this approach is currently limited for ChRmine by the divergence of BCCRs from CCRs and absence of an experimentally determined ChRmine (or any BCCR) structure.

Here, we present cryo-EM structures of ChRmine in apo (opsin) and retinal-bound (rhodopsin) forms to provide insight into the molecular determinants of channel activity in a BCCR, to better understand unique ChRmine properties, and to facilitate rational engineering of ChRmine variants to create new optogenetic tools.

## Results and discussion

We determined cryo-EM structures of ChRmine (residues 1–309 of *Rhodomonas lens* cation channelrhodopsin 1) in lipid nanodiscs (Fig. 1, Table 1). ChRmine was cloned with a cleavable C-terminal GFP tag, expressed in insect cells grown in the presence of 5 μM all-trans retinal, extracted and purified in detergent, and reconstituted into lipid nanodiscs formed by the scaffold protein MSP1E3D1 and a mixture of DOPE, POPC, and POPS lipids (Supplementary Fig. 1).

We observed two populations of ChRmine particles distinguished by nanodisc diameter. Particles with small and large diameter nanodiscs were processed separately and generated reconstructions to 2.7 Å and 3.1 Å resolution, respectively (Fig. 1, Supplementary Figs. 2 and 3; Table 1). ChRmine residues 2–285 were de novo modeled into each map. Residue 2 is best modeled as an acetylalanine, consistent with common posttranslational enzymatic cleavage of the initiating methionine and acetylation of adjacent small hydrophobic residues (Supplementary Fig. 4)[24]. The cytoplasmic C-terminal 19 amino acids are not visible in the cryo-EM map, likely due to disorder. Both maps display clear density for one retinal and multiple lipids per subunit, while the higher resolution map from small diameter ChRmine nanodiscs showed additional density for nineteen ordered water molecules per subunit. As there are no other substantial differences between the two ATR-bound ChRmine structures (overall r.m.s.d. = 0.28 Å), we focus our discussion below on the higher resolution structure unless otherwise noted.

ChRmine is a homotrimer with a central pore between subunits (Fig. 1), unlike all other channelrhodopsins of known structure which

are homodimers[3,25]. The general topology of each ChRmine subunit is typical of rhodopsins, with an extracellular N-terminal region (amino acids 2–26), seven transmembrane helices (TM1-7, amino acids 27–270), intracellular and extracellular linkers (ICL1-3 and ECL1-3), a cytoplasmic C-terminal region (Fig. 1c, d). A conserved lysine (K257) on TM7 is covalently linked to a molecule of all-trans retinal through a Schiff base. Trimerization in ChRmine is achieved through interactions between transmembrane helices, ECL1 linkers, and N-terminal regions (Fig. 1a, b, e). Association of TM1 and TM2 from one subunit with TM4 and TM5 from the neighboring subunit creates a continuous protein interface across the lipid membrane (Fig. 1e). Intersubunit grooves on the extracellular side and a pocket on the intracellular side of the transmembrane interface are occupied by ordered lipid acyl chains (Fig. 1a, b). This arrangement is analogous to ion pump rhodopsins[26,27] and contrasts with CCR homodimers that assemble through interactions between TM3, TM4, and the N-terminal region of each subunit[28–31].

While central pores are also observed in bacteriorhodopsin pump trimers[26], the ChRmine central pore is distinctly shaped with a pair of constrictions separating polar extracellular and nonpolar intracellular central cavities (Fig. 2a). The hourglass-shaped pore is a consequence of uniquely structured ECL1s and N-terminal regions in ChRmine (Fig. 2b). In other channelrhodopsins of known structure, including C1C2[28] and Chrimson[31], ECL1 forms a short two-stranded β-sheet connecting TM2 and TM3 (Fig. 2b). In ChRmine, TM2 is extended by one helical turn and TM3 is partially unwound at its N-terminal end (Fig. 2b). The intervening ECL1 is kinked into a staple-like "C" shape that projects towards the central trimer axis with residues 102–107 lining ~15 Å of the central pore between subunits (Fig. 2b, c). G14-T23 from the N-terminal region forms a short helix and linker that caps ECL1, while A2-V13 extend another ~15 Å into the cytoplasmic solution to form a mouth below the central pore (Fig. 2b, Supplementary Fig. 4).

Could the central pore form a conduction pathway in ChRmine and perhaps help explain its large photocurrents relative to other channelrhodopsins? The extracellular central cavity is indeed wide and electronegative as might be expected for a cation permeation path (Fig. 2e). Acetylated A2, G14, G102, and F108 carbonyls and the D17 side chain line the walls of the extracellular central cavity as it tapers from 5 to 1.4 Å in radius at a polar constriction formed by F104 carbonyls about one-third of the way across the membrane bilayer (Fig. 2a, c, d). However, the intracellular central cavity is nonpolar and packed with lipids that preclude ion passage (Fig. 2a, c, d, e). Hydrophobic side chains from TM2, TM3, and TM4 (including F76, F80, L118, P121, M122, Y125, Y129, A141, F144) line the walls of the intracellular central cavity as it tapers from 6 Å to 1.8 Å in radius at a hydrophobic constriction formed by I106 side chains about two-thirds of the way across the membrane. Consistent with the nonpolar environment (and analogous to ion pump rhodopsin trimers[26,27,32]), we observe strong tube-shaped features in this region that are well fit by three complete and tightly packed DOPE lipid molecules (Fig. 2c, Supplementary Figs. 4, 5). Association of lipid acyl chains results in constrictions less than 1 Å in radius, effectively sealing the pore. The lipids are fenced in the central pore away from bulk lipids due to cytoplasmic extensions of ChRmine ICLs and C-terminal regions that form an intracellular ring connecting the three subunits (Supplementary Fig. 5). Opening a central pore would therefore require dilation of the two central constrictions and breaking intersubunit interactions to permit lipids to diffuse out of the pore. We conclude the central pore is unlikely to serve as a conduction pathway in ChRmine.

Instead, ChRmine displays features consistent with a single conduction pathway within each subunit (Fig. 3). As expected, the channel is captured in a closed ground state conformation, but the relative positions of TM1, TM2, TM3, and TM7 create deep extracellular and intracellular cavities for ion conduction that are only sealed from one another at the Schiff base (Fig. 3a). Comparing the presumed

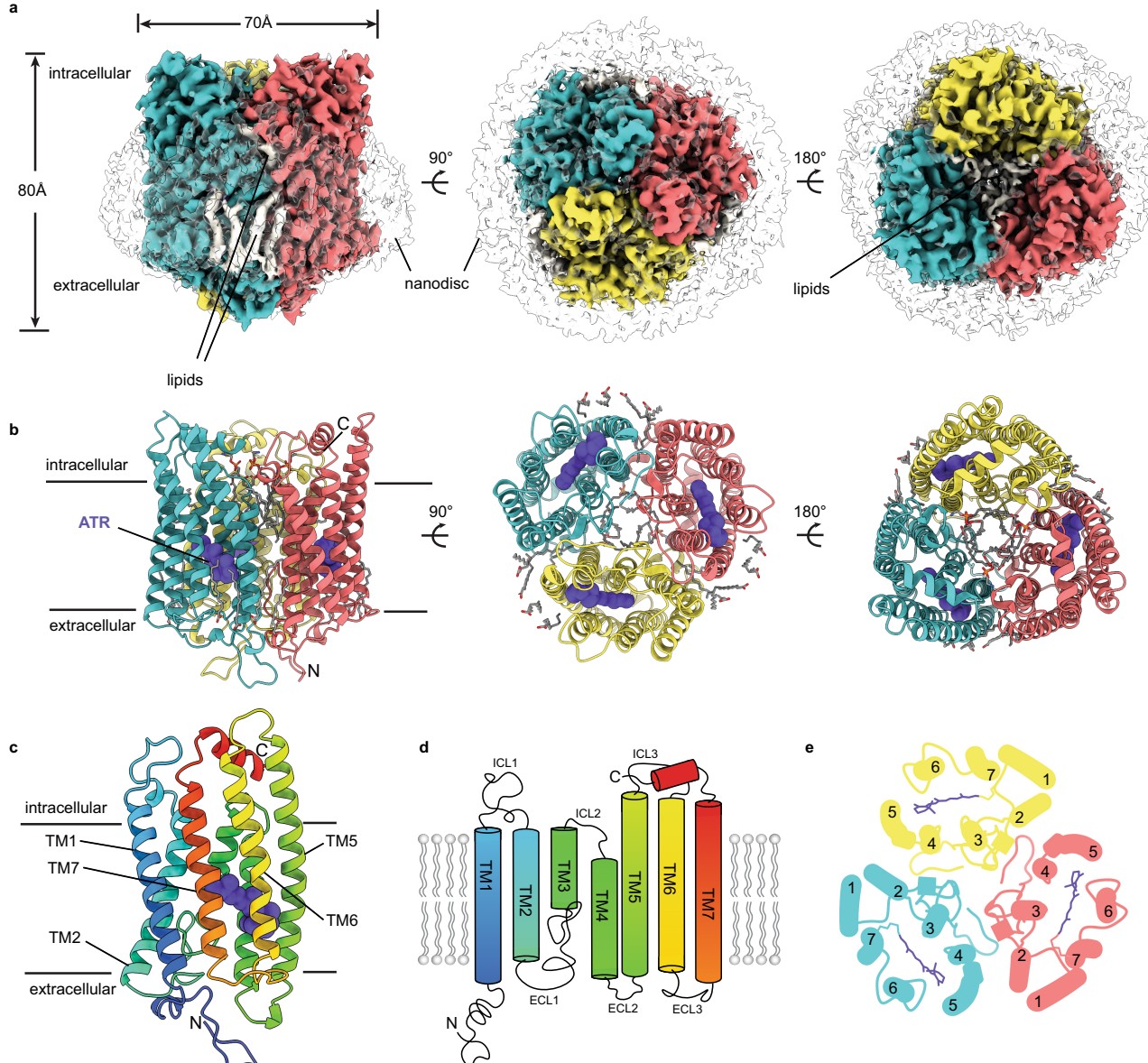

**Fig. 1 | Structure of ChRmine in lipid nanodiscs. a** Cryo-EM map of retinal-bound ChRmine in a MSP1E3D1 nanodisc at 2.7 Å resolution viewed from (left) the membrane plane, (middle) the extracellular side, and (right) the intracellular side. Subunits are colored red, teal, or yellow; lipids are white; and the nanodisc is transparent. **b** Model shown in the same views as (a) with all-trans retinal shown as purple spheres. **c** Model and (**d**) cartoon illustration of a single ChRmine subunit colored blue (N-terminus) to red (C-terminus). **e** Helical arrangement and packing in the ChRmine trimer viewed from the intracellular side and clipped above the all-trans retinal.

conduction pathway in ChRmine to other channelrhodopsins reveals differences on both sides of the membrane that could contribute to properties of ChRmine that differ from other channelrhodopsins.

First, the ChRmine extracellular cavity is wider, deeper, and more electronegative than in other channelrhodopsins including C1C2, Chrimson, and ChR2 (Fig. 3a). The funnel-shaped outer cavity extends from the extracellular solution to K257 and the Schiff base (Fig. 3a–c). Acidic residues including D100, D115, E154, E158, D242, E246, and D253 line the cavity walls to create a highly electronegative environment that favors interactions with cations. The funnel narrows to a single constriction ~3 Å in diameter formed by side chains of D115, D253, Y85, Y116, and K88 just extracellular to the Schiff base (Fig. 3b, c). This contrasts with other channelrhodopsins including C1C2 and Chrimson, which have two narrow extracellular tunnels that terminate ~5 Å below the Schiff base[28,31]. The differences in diameter are predominantly due to changes at three sites that correspond to the central and extracellular gates of other

channelrhodopsins. First, the central gate in C1C2 (that involves S102, E129 ("E3"), and N297) is absent in ChRmine due to smaller analogous side chains (L40, A81, and S258) and differences in their relative position(Fig. 3b). Second, an outer gate constriction in C1C2 (that involves K132 and E136 ("E4")) is structured differently in ChRmine due to smaller side chains (S84 and K88) and kinking of the adjacent TM3, which together open a larger path towards the Schiff base(Fig. 3c). Third, an additional outer gate constriction in C1C2 (that involves E140 ("E5") and T285) is dramatically dilated in ChRmine due to extension and displacement of TM2 and rearrangement of the TM2-TM3 linker. ChRmine's D92 is positioned ~6 Å further away from the conduction path than the analogous residue C1C2 E140, while electronegativity of the wider ChRmine cavity is increased by the substitution of E246 in place of C1C2 T285(Fig. 3c).

The ChRmine intracellular cavity is also larger, deeper, and more forked relative to other channelrhodopsins (Fig. 3a). In ChRmine, three openings at the intracellular surface (between TM2 and TM3, between

**Table 1 | Cryo-EM data collection, refinement, and validation statistics**

| | ChRmine small nanodisc (PDB 7SFK) (EMDB 25091) | ChRmine large nanodisc (PDB 7SFJ) (EMDB 25079) | apo-ChRmine nanodisc (PDB 7SHS) (EMDB 25135) |
|---|---|---|---|
| **Data collection and processing** | | | |
| Magnification | ×36,000 | ×36,000 | ×36,000 |
| Voltage (kV) | 200 | 200 | 200 |
| Electron exposure (e⁻/Å²) | 50 | 50 | 50 |
| Defocus range (μm) | −0.6 to −2.0 | −0.6 to −2.0 | −0.6 to −2.0 |
| Super resolution pixel size (Å) | 0.5685 | 0.5685 | 0.5685 |
| Binned pixel size (Å) | 1.137 | 1.137 | 1.137 |
| Symmetry imposed | C3 | C3 | C1 |
| Initial particle images (no.) | 8,444,523 | 8,444,523 | 7,054,805 |
| Final particle images (no.) | 81,839 | 100,946 | 41,053 |
| Map resolution (Å) | 2.7 | 3.1 | 4.1 |
| FSC threshold | 0.143 | 0.143 | 0.143 |
| **Refinement** | | | |
| Initial model used (PDB code) | de novo | de novo | 7SFK |
| Model resolution (Å) | 2.7 | 3.0 | 4.1 |
| FSC threshold | 0.143 | 0.143 | 0.143 |
| Model composition | | | |
| Nonhydrogen atoms | 7419 | 7377 | 6873 |
| Protein residues | 852 | 852 | 849 |
| Ligands | 24 | 24 | 0 |
| Waters | 57 | 3 | 0 |
| Mean *B* factors (Å²) | | | |
| Protein | 14.21 | 59.86 | 149.44 |
| Ligand | 20.21 | 58.10 | – |
| Water | 16.05 | 51.81 | – |
| R.m.s. deviations | | | |
| Bond lengths (Å) | 0.003 | 0.003 | 0.003 |
| Bond angles (°) | 0.536 | 0.537 | 0.740 |
| CC mask | 0.85 | 0.79 | 0.70 |
| **Validation** | | | |
| MolProbity score | 1.15 | 1.27 | 1.33 |
| Clashscore | 3.61 | 3.87 | 5.95 |
| Poor rotamers (%) | 0.43 | 0.85 | 0 |
| Ramachandran plot | | | |
| Favored (%) | 98.22 | 97.51 | 98.22 |
| Allowed (%) | 1.78 | 2.49 | 1.78 |
| Disallowed (%) | 0 | 0 | 0 |

TM1 and TM7, and above TM2) merge into a single cavity. Acidic residues including E50, E70, D126, and E280 create an electronegative surface that is expected to favor interactions with cations (Fig. 3d, e). The cavity extends deep into the channel before terminating just intracellular to K257 and the Schiff base. This contrasts with other channelrhodopsins including C1C2, Chrimson, and ChR2 which have a shorter intracellular cavity terminating at an inner gate ~7 Å closer to the intracellular solution. The difference is predominantly due to rearrangement of the region corresponding to the intracellular gate in other channelrhodopsins. In C1C2, the intracellular gate involves Y109, E121 ("E1"), E122 ("E2"), and H173 and creates a tight seal at the end of the intracellular cavity. In ChRmine, the smaller corresponding

residues L47, L73, A74, and D126 constrict the cavity to a radius of ~1.5 Å, but do not form a tight seal (Fig. 3a, d, e).

We next consider the structure around the retinal chromophore to better understand spectral and kinetic properties of ChRmine. We focus on three aspects of this region that are unique to ChRmine and have been shown to impact light absorption and channel closing rates in other channelrhodopsins: (i) electrostatic interactions around the Schiff base, (ii) polarity around the retinal β-ionone, and (iii) packing around the retinal polyene and β-ionone moieties.

Electrostatic interactions around the protonated Schiff base are determinants of rhodopsin spectral properties[2–6]. Generally, negative charge provided by the protein (a counterion) stabilizes the protonated ground state and results in blue light absorption, while the absence of negative charge or increased distance between negative charge and the Schiff base destabilizes the ground state and results in red shifted absorption. A negatively charged group typically also acts as a proton acceptor during the photocycle. ChRmine has negatively charged groups extracellular to the Schiff base like other rhodopsins, but their positioning and interaction network is distinct (Fig. 4a). In ChRmine, D115 and D253 correspond to negative charges in other channel (E162 and D292 in C1C2) and pump rhodopsins (D85 and D212 in HsBr). In C1C2, D292 acts as the proton acceptor and E162 is not required for channel function[28,33] (Fig. 4b). In HsBr, D85 acts as the proton acceptor and D212 is positioned by hydrogen bonds with Y57 and Y185[34] (Fig. 4c). ChRmine D253 is held by hydrogen bonds with two tyrosines (Y85 and Y116) like HsBr D212, although the second tyrosine is contributed by TM6 in HsBR and TM3 in ChRmine (Fig. 4a, c). However, due to bending and unwinding of the N-terminal end of TM3 in ChRmine, D115 is further away from the Schiff base (7.0 Å) and rather forms an interaction through a water molecule positioned where the side chain carboxylates C1C2 E162 or HsBr D85 are found (Fig. 4a, b, d). This charge separation could contribute to red-shifted spectra of ChRmine.

A second determinant of rhodopsin spectral characteristics is the polarity around the retinal β-ionone ring. Electronegative binding pockets generally result in red-shifted absorption spectra, perhaps due to stabilization of a retinal isomerization intermediate in which charge is transferred towards the β-ionone ring[31,35]. In ChRmine, the pocket around the β-ionone ring is distinctly electronegative due to contributions from S149, E154, and Y113 residues (Fig. 5a–c). The red-shifted channlerhodopsin Chrimson has a comparably electronegative β-ionone pocket due to similarly positioned serine (S223) contributed from TM5 (compared to S149 contributed from TM4 in ChRmine)[31]. In the relatively blue-shifted C1C2, however, two of three corresponding residues (W201 and A206) are nonpolar or positioned further away from the retinal, resulting in a less electronegative retinal binding pocket[28,31].

The shape of the retinal binding pocket has also been shown to influence closing rate of other channelrhodopsins. Tight packing around the retinal polyene and β-ionone ring is correlated with fast closure, while mutations that loosen packing have been shown to slow closing, presumably by impairing coupling between retinal and protein conformational changes[31,36]. The retinal binding pocket is substantially less tightly packed on three sides compared to C1C2 or Chrimson, consistent with the 5-10-fold slower closing rate ChRmine. The intracellular and extracellular sides of the retinal binding pocket are formed by aromatic residues in all three opsins. In C1C2 and Chrimson, the polyene chain is tightly sandwiched between tryptophans (W163/W262 and W166/W265, respectively). In ChRmine, the extracellular tryptophan is substituted by a smaller tyrosine (Y116) and the intracellular tryptophan (W223) is displaced away from the retinal due to kinking of TM6 (Fig. 5a–c). Along the lateral axis of the pocket facing TM3 and TM4, M201 in Chrimson closely abuts the retinal and its mutation to smaller residues has been shown to decrease closing kinetics[31]. In ChRmine, I146 in this position does not extend as far along the polyene

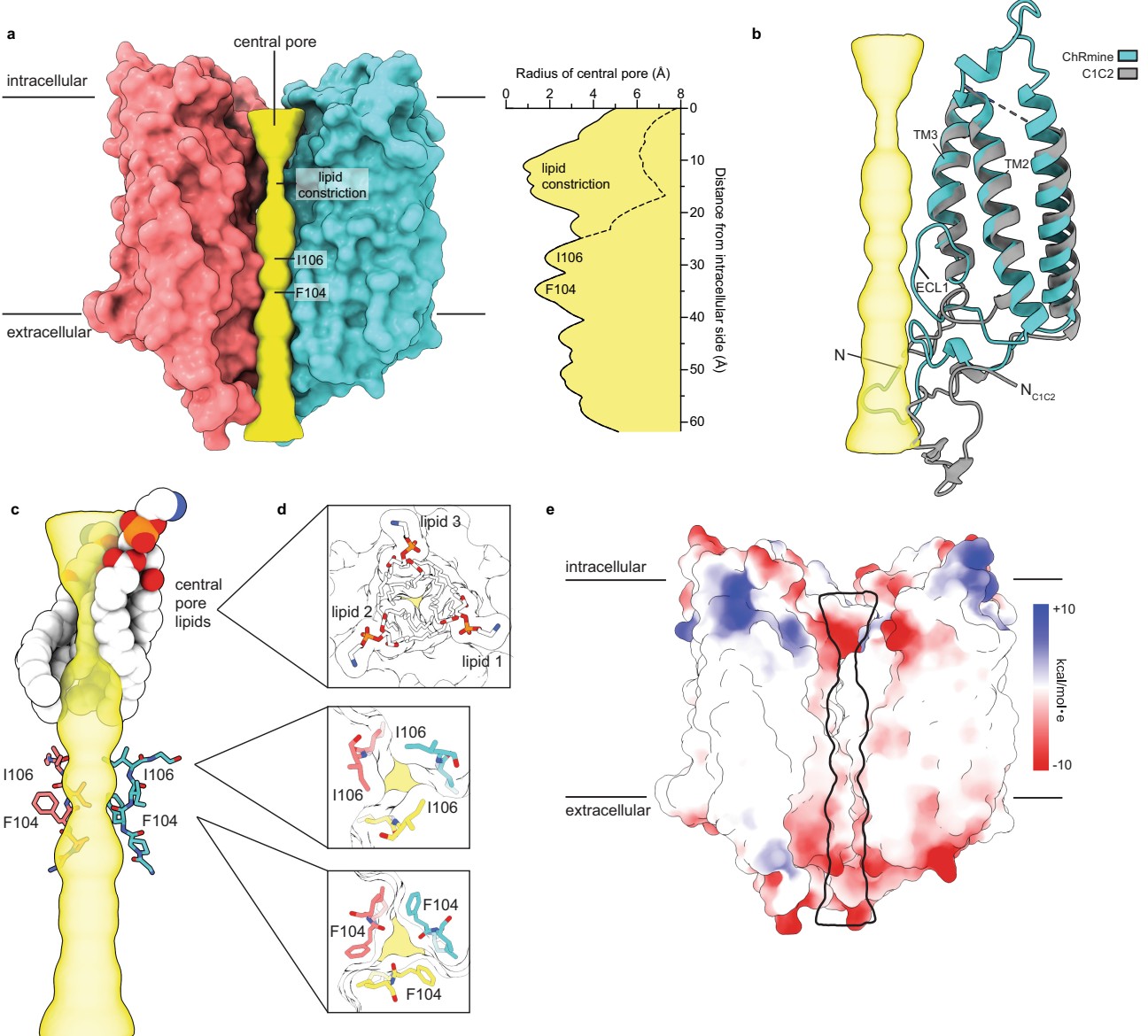

**Fig. 2 | A lipid-filled central pore in ChRmine. a** View from the membrane plane of the ChRmine central pore (yellow) against the surface of two channel subunits (the front subunit is not displayed). Corresponding pore radius as a function of distance from the intracellular side is shown on the same scale. **b** Overlay of a ChRmine and a C1C2 subunit. The uniquely structured ECL1 and N-terminal region of ChRmine line the central pore. **c** Constrictions of the central pore formed by lipids and ECL1 viewed from the membrane plane and **d** from the intracellular side. **e** Electrostatic character of the central pore drawn on the ChRmine surface in the same view as (**a**).

ring (Fig. 5d). The consequence of these three differences in the ChRmine retinal binding pocket are large gaps on three sides of the retinal between C8-C13 not observed in other channelrhodopsins.

During preparation of this manuscript, an independent report of a retinal-bound ChRmine structure determined in detergent micelles and in complex with antibody fragments to 2.0 Å resolution was posted and published[37]. Two substantial differences are observed between that structure and the ones reported here. The first is around the extracellular N-terminal region and does not have obvious functional consequences. We expressed ChRmine without N-terminal modifications and found the N-terminus from each subunit forms a closed ring at the extracellular side of the central pore. This region is not observed in the detergent-solubilized structure[37], probably because additional residues from an N-terminal epitope tag sterically clash with the N-terminal ring, resulting in local disorder.

The second difference is in the central pore and has major consequences for functional interpretation. We determined

ChRmine structures in lipid nanodiscs and observed tightly packed lipids in the hydrophobic intracellular cavity of the central pore that preclude ion conduction. This observation is consistent with structures of trimeric ion pump rhodopsins that have similar lipid-filled central pores[26]. In contrast, lipids are not modeled in the central pore of the detergent solubilized structure[37]. The most likely reason for this difference is that centrally bound lipids are extracted or replaced by less ordered detergent molecules when ChRmine is solubilized in detergent micelles. In the absence of centrally bound lipids, the central pore was predicted to form an ion conduction path that could open upon retinal isomerization and dilation of the F104 and I106 constrictions[37]. We conclude central conduction in trimeric ChRmine is unlikely for several reasons: (1) similarity to other trimeric pump rhodopsins, (2) the hydrophobic nature of the pore, and (3) the presence of clearly resolved lipids in the structure of ChRmine reconstituted into lipid nanodiscs reported here.

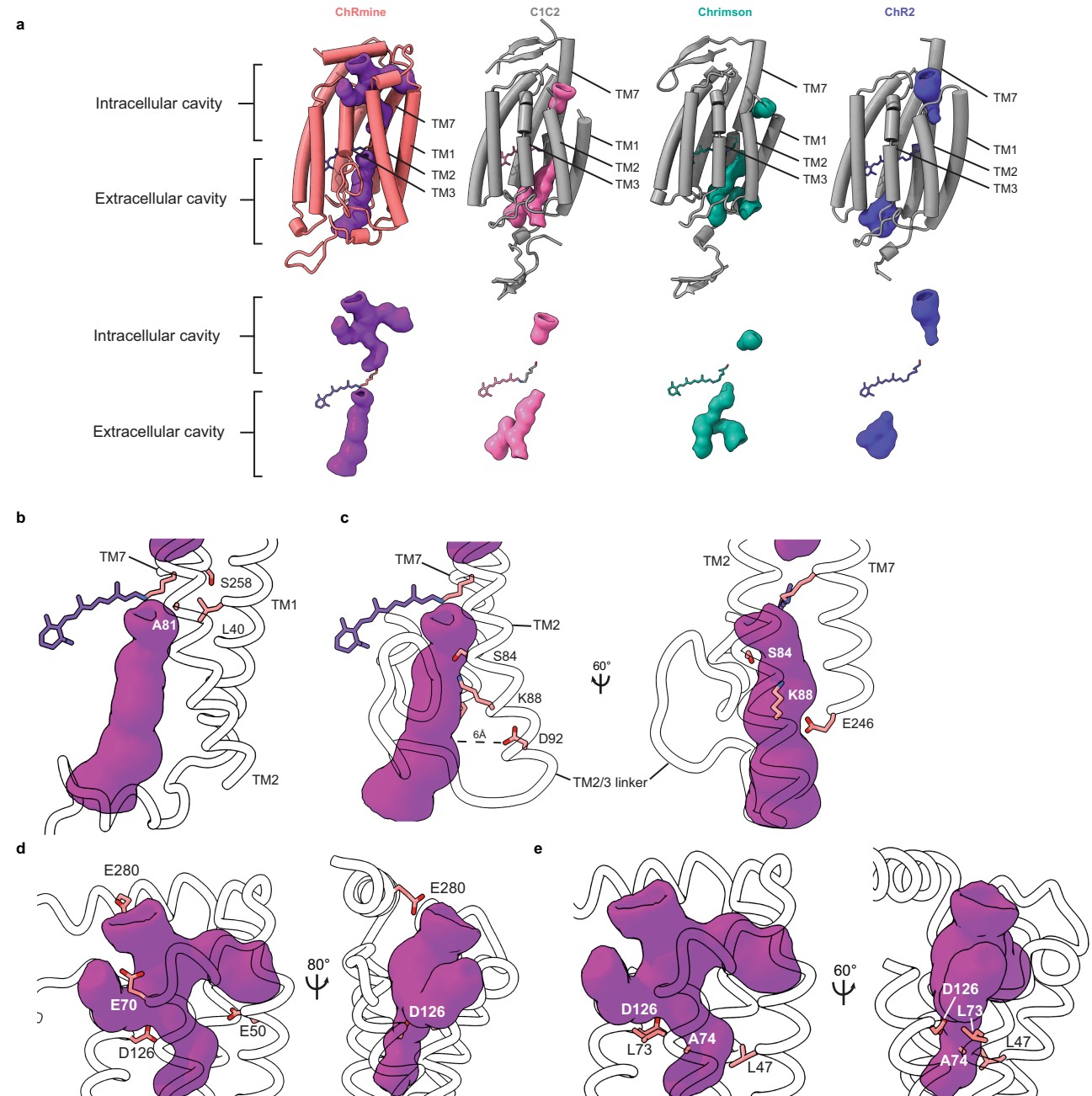

**Fig. 3 | The ChRmine ion conduction pathway. a** Ion conducting pathways in ChRmine, C1C2, Chrimson, and ChR2. A single subunit is shown with helices as tubes (upper). Surfaces of ion conducting pathways are purple, pink, green, or blue. Zoomed in views of the ChRmine (**b**, **c**) extracellular and **d**, **e** intracellular cavities with key residues indicated.

We generated mutations in ChRmine to test the functional impact of residues implicated as potential determinants of channel properties (Fig. 4e–i). We reasoned mutations at Y116 would decelerate ChRmine closing because Y116 hydrogen bonds with D253 in the counterion network, forms part of the floor of the retinal binding pocket, and contributes to the extracellular constriction. Indeed, Y116M or Y116W mutants that disrupt hydrogen bonding and packing interactions slow closing ~10-fold (Fig. 4i). Both mutants also reduce photocurrents (~4- and 50-fold compared to wild-type ChRmine, respectively), though we cannot rule differences in expression level from contributing to this effect (Fig. 4g).

We then reasoned other manipulations to the counterion network or conduction path could have opposing effects and accelerate ChRmine closing. S169 in Chrimson contacts the Schiff base and an S169A mutant accelerates channel closing[31]. Consistently, we found the analogous T119A mutation in ChRmine accelerated channel closing ~2-fold without significantly impacting photocurrents or light sensitivity relative to wild-type (Fig. 4e–i). We confirmed that the reported mutation H33R (called "high speed" or hsChRmine)[37], which lines the extracellular cavity, also accelerated closing. In contrast to T119A though, H33R showed significantly decreased photocurrents and light sensitivity (Fig. 4g, h). ChRmine T119A could therefore be preferable to wild-type or H33R ChRmine for optogenetic applications.

We next tested mutations that alter retinal binding pocket properties. A ChRmine I146M/G174S mutation (called "red-shifted" or rsChRmine) was reported to alter spectral properties[37]. We reasoned it would also accelerate closing rate because the mutations result in a tighter retinal binding pocket: I146 is analogous to M201 in Chrimson[31]

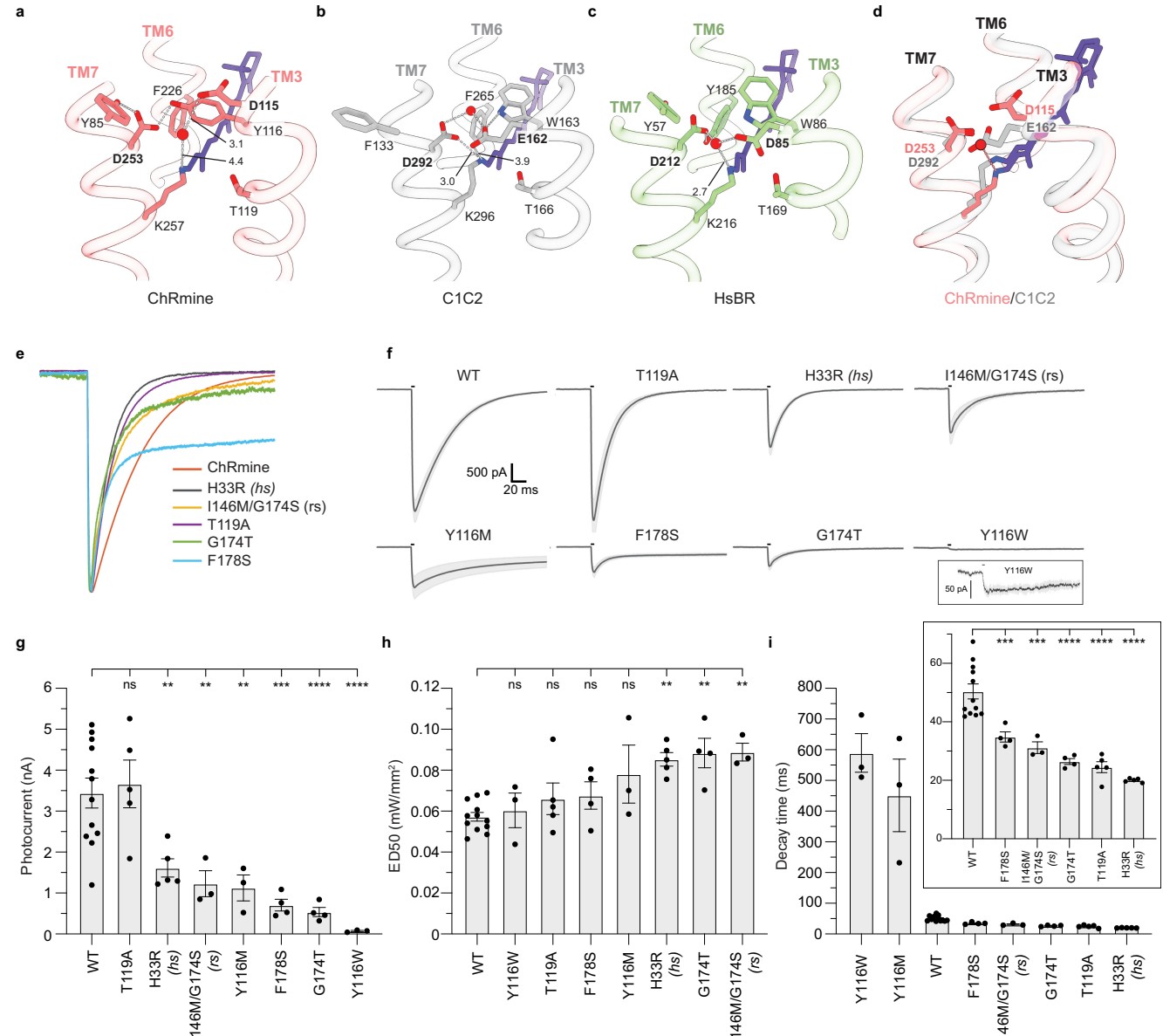

**Fig. 4 | Schiff base chemistry and function of ChRmine variants.** Views of the area around the protonated Schiff base in **a** ChRmine, (**b**) C1C2, (**c**) HsBr, and **d** ChRmine and C1C2 overlaid. Negatively charged counterions and proton acceptors, interacting residues, and key water molecules are indicated. **e** Peak-normalized currents for ChRmine and five mutants with faster closing rate. Note long-lived F178S steady-state current. **f** Average photocurrents for ChRmine and each mutant evaluated. The bar above each trace indicates a 5 ms light pulse. Y116W is shown in inset with ten-fold expanded current scale. **g** Peak photocurrents, (**h**) light sensitivity, and **i** decay time constant for each channel. Values are mean ± sem

for $n = 12$, 5, 5, 3, 3, 4, 4, and 3 cells for wild-type ChRmine, T119A, H33R, I146M/G174S, Y116M, F178S, G174T, and Y116W, respectively. Differences assessed with one-way analysis of variance (ANOVA) with Dunnett correction for multiple comparisons (ns, not significant ($P = 0.9994$ for T119A in (**i**) and 0.9995, 0.7446, 0.6626, and 0.0957 for Y116W, T119A, F178S, and Y116M in **h**, respectively); **$P < 0.01$; ***$P < 0.001$; ****$P < 0.0001$). **i**, inset Expanded scale comparing wild-type ChRmine and mutants with accelerated closing rate. Source data for **g**–**i** are provided as a Source Data file.

that packs against the retinal face and a gap exists between G174 and the retinal indole. Indeed, I146M/G174S ChRmine closed 1.6-fold faster than wild-type. We generated additional mutations predicted to either directly (G174T) or indirectly (F178S, by making space for Y182 to pack against the retinal as seen in all other channelrhodopsin structures) create a tighter retinal pocket. Consistently, G174T and F178S ChRmine displayed accelerated closing (1.9- and 1.4-fold), though we note F178S closing is biphasic with a second slower step.

To date, there are no channelrhodopsin structures without bound retinal and the structural consequences of retinal binding, which could have implications for the design of functional variants, are unknown. We purified apo-ChRmine from cells grown without addition of all-

trans retinal, reconstituted it into lipid nanodiscs, and determined its structure to 4.1 Å resolution (Supplementary Fig. 6, Table 1). The lower resolution of the apo structure is due in part to fewer particles and anisotropy of the reconstruction due to preferred particle orientation and potentially an increase in overall ChRmine flexibility. The apo-ChRmine structure (consisting of residues 3–285) was modeled by combining portions de novo built (into well-defined regions) with docked portions of the ATR-bound structure (into less well-defined regions) prior to structure refinement. The structures are similar overall (r.m.s.d. = 0.8 Å); no major rearrangements in the overall fold are observed (Supplementary Fig. 7). However, subtle conformational changes are focused to the intracellular halves of TM1, TM5, TM6, and

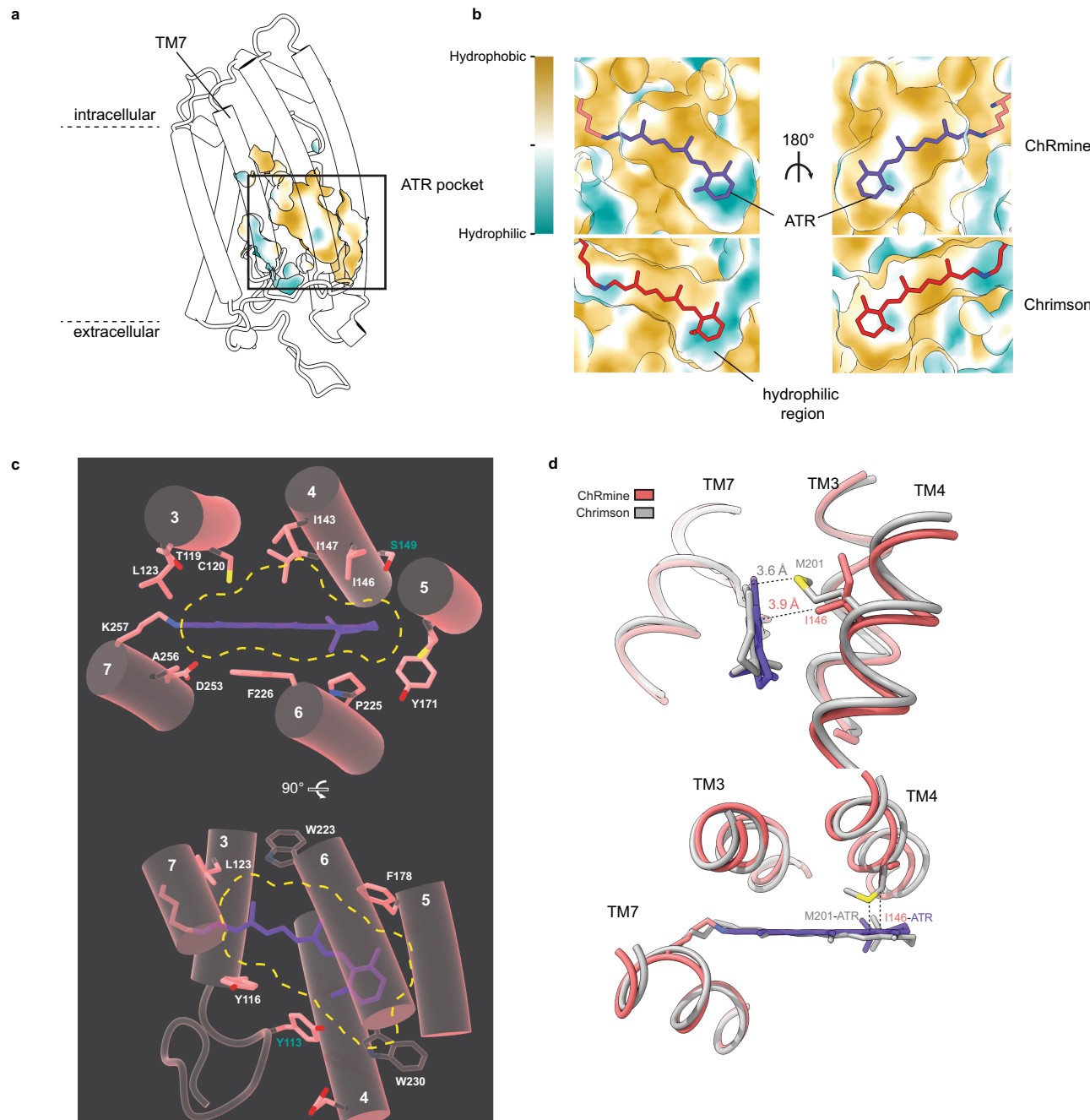

**Fig. 5 | The ChRmine retinal binding pocket. a** ChRmine viewed from the membrane plane with position of retinal binding pocket highlighted. **b** Views of the retinal binding pockets of (top) ChRmine and (bottom) Chrimson. Protein surface is colored from hydrophobic (brown) to hydrophilic (teal). **c** ChRmine retinal binding pocket with key residues drawn as sticks. **d** Overlay of ChRmine and Chrimson illustrating differences in packing environment.

TM7, which bow out toward the membrane ~0.5 Å, 1 Å, 2 Å, and 1 Å, respectively (Supplementary Fig. 7). The overall effect is to expand the retinal binding pocket and loosen packing between TM6 and its neighboring helices TM5 and TM7 on the membrane facing sides. These changes may facilitate loading apo-ChRmine with retinal.

Together, our results provide insight into the unique structure and functional properties of ChRmine and suggest design principles for new and improved channelrhodopsins. ChRmine displays large photocurrents compared to other channelrhodopsins when expressed in neurons or cultured cells[13,17]. It is unknown how differences in single channel conductance and/or expression levels contribute to large photocurrents for ChRmine compared to other CCRs. *R. lens* CCR1 (from which ChRmine is derived) does not show substantially larger

photocurrents compared to other BCCRs[38] in HEK293 cells, suggesting some contribution of expression level differences. We speculate large photocurrents in neurons could be due, at least in part, to differences along the canonical conduction pathway within each subunit that expand and deepen cavities on either side of the membrane. This would be consistent with hsChRmine (H33R)[37], which is predicted to constrict the extracellular pathway, reducing photocurrents ~2-fold in addition to accelerating closing rate (Fig. 4). We further speculate that chemistry around the Schiff base, counterion network structure, and a loosely-packed retinal binding pocket contribute to slow closing in ChRmine. Mutations that markedly slow (Y116W, Y116M) or accelerate closing rate (T119A, G174T, F178S, and I146M/G174S (rsChRmine[37])) support this hypothesis.

We identify one mutation, ChRmine T119A, with properties of a promising optogenetic tool. ChRmine T119A retains large photocurrents and high sensitivity of the wild-type channel while accelerating closing rate two-fold to ~25 ms. Further work to evaluate models discussed above with targeted mutations are likely to result in additional variants that maintain favorable properties of ChRmine while narrowing its absorbance spectrum and accelerating closing rate. Such tools could enable new experiments including large-scale or deep-tissue optogenetics and manipulations in challenging models such as primates.

## Methods

### Cloning and protein expression

The coding sequence for ChRmine from *Rhodomonas lens* was cloned into a custom vector based on the pACEBAC1 backbone (MultiBac; Geneva Biotech, Geneva, Switzerland) with an added C-terminal Pre-Scission protease (PPX) cleavage site, linker sequence, superfolder GFP (sfGFP), and 7xHis tag to generate the construct ChRmine-LNS-LEVLFQGP-SRGGSGAAAGSGSGS-sfGFP-GSS-7xHis for expression in insect cells. Primers used to clone ChRmine into the custom vector:

Forward- AATGATACGGCGACCACCGATctcgagACCACCATGGCACAC,

Reverse- tcGAATTCagAGACAGTCTCCGCAG.

Mutations were introduced using inverse PCR. The following primers were used to generate mutants: T119Afwd TGCTGGCCTGCCCAATGCTG, T119Arev GGCAGGCCAGCATGTAGTCGG, I146Mfwd TTTGCCATGCTGATGTCTGGCGTG, I146Mrev GACATCAGCATGGCAAAGATGATGGCGC, G174Sfwd CGCCTGGTATGGCTTTAGCTGTTTCTGGTTTATCTTCGCCTACTCTATCGTGATGAGC, G174Srev CAGAAACAGCTAAAGCCATACCAGGCGTAGGCGCCATTCCTCAGCCTAG, G174Tfwd GCTTTACCTGTTTCTGGTTTATCTTCGC, G174Trev AAACAGGTAAAGCCATACCAGGC, F178Sfwd TTCTGGTCCATCTTCGCCTACTCTATCGTG, F178Srev GAAGATGGACCAGAAACAGCCAAAGCCATAC, Y116Wfwd GACTGGATGCTGACCTGCCCAATG, Y116Wrev GGTCAGCATCCAGTCGGCATACCT, Y116Mfwd GACATGATGCTGACCTGCCCAATG, Y116Mrev GGTCAGCATCATGTCGGCATACCT, H33Rfwd GGCGCCAGGTGGTCTTGCTTTATCGTG, H33Rrev AGACCACCTGGCGCCGATGGC.

The ChRmine construct was transformed into DH10Bac *E. coli* to generate a bacmid according to the manufacturer's instructions. Subsequent steps used *Spodopetera frugiperda* SF9 cells cultured in ESF 921 medium (Expression Systems, Davis, CA). Bacmid was transfected into adherent SF9 cells using the Escort IV (MilliporeSigma, Burlington, MA) to produce P1 virus. P1 virus was used to infect SF9 cells in suspension at 2 million cells/mL at a multiplicity of infection (MOI) ~0.1 to generate P2 virus. Infection was monitored by fluorescence and P2 virus was harvested 48–72 hours post infection. P3 virus was generated in a similar manner. P3 viral stock was then used to infect Sf9 cells at 2–4 million cells/mL at a MOI ~2–5 for large scale protein expression. For ATR-bound ChRmine samples, 5 μM all-trans retinal (ATR) was added to media 48 hours post infection and cells continued to grow for an additional 12–16 h. Cells were harvested by centrifugation at $5000 \times g$ for 15 min, flash-frozen in liquid nitrogen, and stored at −80 °C.

### Protein purification

Cells from 0.5–1 L of culture (~7.5–15 mL cell pellet) were thawed and resuspended in 100 mL of lysis buffer (50 mM Tris, 150 mM NaCl, 1 mM EDTA pH 8). Protease inhibitors (final concentrations: E64 (1 μM), Pepstatin A (1 μg/mL), Soy Trypsin Inhibitor (10 μg/mL), Benzimidine (1 mM), Aprotinin (1 μg/mL), Leupeptin (1 μg/mL), AEBSF (1 mM), and PMSF (1 mM)) were added immediately before use and Benzonase (4 μl) was added after the cell pellet thawed. Cells were lysed by sonication and membranes pelleted by centrifugation at 150,000 x g for 45 minutes. The supernatant was discarded and membrane pellets were scooped into a dounce homogenizer containing extraction buffer

(50 mM Tris, 150 mM NaCl, 1 mM EDTA, 1% n-Dodecyl-β-D-Maltopyranoside (DDM, Anatrace, Maumee, OH), pH 8). A 10% solution of DDM was dissolved and clarified by bath sonication in 200 mM Tris pH 8 prior to addition to buffer to the indicated final concentration. Membrane pellets were homogenized in extraction buffer and this mixture (150 mL final volume) was gently stirred at 4 °C for 1–2 hr. The extraction mixture was centrifuged at 33,000 x g for 45 minutes and the supernatant, containing solubilized ChRmine, was bound to 5 mL of Sepharose resin coupled to anti-GFP nanobody for 1–2 hour at 4 °C. The resin was collected in a column and washed with 10 mL of buffer 1 (20 mM Tris, 150 mM NaCl, 1 mM EDTA, 0.025% DDM, pH 7.4), 40 mL of buffer 2 (20 mM Tris, 150 mM NaCl, 0.025% DDM, pH 7.5), and 10 mL of buffer 1. The resin was then resuspended in 6 mL of buffer 1 with 0.5 mg of PPX and rocked gently in a concial tube overnight at 4 °C. Cleaved ChRmine protein was then eluted with an additional 8 mL of buffer 1, spin concentrated to 500 μl with a 10 kDa cutoff Amicon Ultra spin concentrator (Millipore), and then loaded onto a Superose 6 increase column (GE Healthcare, Chicago, IL) on an NGC system (Bio-Rad, Hercules, CA) equilibrated in buffer 3 (20 mM Tris, 150 mM NaCl, 0.025% DDM, pH 7.4). Peak fractions containing ChRmine were collected and spin concentrated prior to incorporation into MSP1E3D1 nanodiscs.

### Nanodisc reconstitution

Freshly purified ChRmine in buffer 3 was reconstituted into nanodiscs formed by the scaffold protein MSP1E3D1 and a mixture of lipids (DOPE:POPS:POPC at a 2:1:1 molar ratio, Avanti, Alabaster, Alabama) at a final molar ratio of 1:4:400 (monomer ratio: ChRmine, MSP1E3D1, lipid mixture). Final concentrations were 20uM ChRmine, 80uM MSP1E3D1, 8 mM lipid mix, and 5 mM DDM in buffer (20 mM Tris pH 7.5, 150 mM NaCl). Lipids solubilized in buffer (20 mM Tris, 150 mM NaCl, pH 7.5) and DDM detergent were first added to ChRmine protein and incubated at 4 °C for 30 minutes. Purified MSP1E3D1 (prepared as described without His-tag cleavage) was then added and the solution was mixed at 4 °C for 10 min before addition of 100 mg of Biobeads SM2 (Bio-Rad). Biobeads were washed into methanol, water, and then buffer and weighed after liquid was removed by a P1000 tip prior to use. This final mixture was gently tumbled at 4 °C for ~12 h. Supernatant was cleared of beads by letting large beads settle and carefully removing liquid with a pipette. The sample was spun for 10 minutes at 21,000 x g before loading onto a Superose 6 increase column in 20 mM Tris, 150 mM NaCl, pH 7.5. Peak fractions corresponding to ChRmine protein in MSP1E3D1 were collected for grid preparation.

### Cryo-EM grid preparation

ChRmine in nanosdiscs was concentrated (10 kDa cutoff) to 2.5 or 14.4 mg ml$^{-1}$ for apo and ATR-bound samples, respectively, and cleared by centrifugation at $21,000 \times g$ for 15 min at 4 °C prior to grid preparation. A 3-μl drop of protein was applied to freshly glow discharged Holey Carbon, 300 mesh R 1.2/1.3 gold grids (Quantifoil). Samples were plunge frozen in liquid nitrogen cooled liquid ethane using a FEI Vitrobot Mark IV (Thermo Fisher Scientific) at 4 °C, 100% humidity, 1 blot force, ~5 s wait time, and 3 s blot time.

### Cryo-EM data acquisition

Grids were clipped and transferred to a 200 kV Talos Arctica microscope. Fifty frame videos were recorded on a Gatan K3 Summit direct electron detector in super-resolution counting mode with pixel size of 0.5685 Å. The electron dose was 1 e$^{-}$ (Å$^2$ s)$^{-1}$ and total dose was 50 e$^{-}$ (Å$^2$)$^{-1}$. Nine movies were collected around a central hole position with image shift and defocus was varied from −0.6 to −2.0 μm through SerialEM[39]. Apo-chRmine was collected with a 0.4um offset from hole center due to higher particle distribution along the carbon edge. See Table 1 for data collection statistics.

## Cryo-EM data processing

For ATR-bound ChRmine, motion correction and dose weighting were performed on 10,900 movies using RELION 3.1's implementation of MotionCor2 and twice 'binned' to 1.137 Å per pixel[40–42]. Contrast transfer function parameters were fit with CTFFIND 4.1[43]. 8,444,523 particles were template-based autopicked from movies CTF fit to 3.7 Å or better, extracted at a 220 pixel box size, and transferred to cryoSPARC v3[44,45]. After iterative rounds of 2D classification and three class ab initio jobs, two distinct ChRmine particle sets emerged with 192,008 particles in large diameter nanodiscs and 162,477 particles in small diameter nanodiscs. The large and small diameter particle sets were refined separately using non-uniform refinement to a nominal resolution of 3.9 Å and 3.1 Å, respectively.

The small diameter nanodisc particle set was transferred to RELION3.1 for initial Bayesian polishing followed by CTF refinement and a second round of Bayesian polishing. Particles were then imported into cryosparc and subjected to five additional rounds of 2D classification resulting in a final set of 81,839 particles. An ab initio reconstruction was performed to provide an initial volume and subsequent non-uniform refinement with C3 symmetry imposed resulted in a final map at 2.7 Å nominal resolution.

The large nanodisc particle set was further 2D classified for three additional rounds to remove junk, low quality, and damaged particles. This resulted in a final set of 105,606 particles which were subjected to ab initio reconstruction and non-uniform refinement with C3 symmetry imposed to generate a 3.2 Å resolution map. These particles were then transferred back to RELION3.1 for two rounds of CTF refinement and Bayesian polishing before being re-imported into cryoSPARC and two-dimensionally classified. Ab initio reconstruction and subsequent non-uniform refinement with C3 symmetry imposed on a final 100,946 particle set resulted in a final map at 3.1 Å nominal resolution.

For apo-ChRmine, patch motion correction was performed on 5339 movies using cryoSPARC v2 and later twice 'binned' to 1.137 Å per pixel. Contrast transfer function parameters were estimated with patch-based CTF and used to eliminate poor quality data, resulting in 3704 movies. Particles were automatically picked using cryoSPARC's blob picker and the resulting good classes were used for the template-based autopicking of 4,431,227 particles. These particles were extracted at a box size of 220 pixels and two-dimensionally classified yielding 260,305 particles. Subsequently, particles were primarily two-dimensionally classified with the exception of three ab initio jobs using five and four classes. Iterative classification resulted in 41,053 particles. After iterative rounds of two- and three- dimension classification a final set of particles were used to non-uniform refinement to achieve a resolution of 4.1 Å.

UCSF pyem tools were used to convert data from cryoSPARC to RELION format.

## Modeling, refinement, and analysis

Cryo-EM maps were sharpened in cryoSPARC and used to build models de novo in Coot[46]. Models were refined in Phenix[47] and assessed for proper stereochemistry and geometry using Molprobity[48]. Structural analysis and figure preparation were performed with ChimeraX[49], HOLE[50], DALI[51], PyMOL, JalView[52], Prism 8, and Adobe Photoshop and Illustrator software.

## Electrophysiology

Chinese hamster ovary (CHO) cells were transfected with either wild-type or mutant ChRmine fused to surface expression (SE) and endoplasmic-reticulum export (ER) motifs and tagged with the fluorophore mRuby2 fused to the proximal clustering, soma-targeting (ST) domain of the potassium channel Kv2.1(SE-ChRmine-mRuby2-ST-ER). 1–2 days after transfection, photostimulation of cells was performed at 510 nm using a Spectra X light engine (Lumencor). Patch pipettes of 4–6 MOhm resistance were pulled from borosilicate glass filaments (Sutter Instruments) and filled with K-gluconate solution (in mM: 110 K-gluconate, 10 HEPES, 1 EGTA, 20 KCl, 2 $MgCl_2$, 2 $Na_2ATP$, 0.25 $Na_3GTP$, 10 Phosphocreatine, 295 mOsm, pH 7.45). Data was recorded at 20 kHz using 700b Multiclamp Axon Amplifier (Molecular Devices). Bath solution (in mM: 119 NaCl, 2.5 KCl, 1.3 MgSO4, 1.3 $NaH_2PO_4$, 20 glucose, 26 NaHCO3, 2.5 $CaCl_2$) was maintained at 30–32° with inline heating. 5–10 μM all-trans retinal was added prior to use. All data was acquired and analyzed with custom code written in Matlab. Currents were measured at a holding potential of −60 mV. Decay time constants were measured by fitting the traces from stimuli offset to a single exponential. Light sensitivity is reported as ED50 from a five-parameter logistic regression to plots of photocurrent versus measured light intensity recorded from successive sweeps with varying LED power[53].

## Reporting summary

Further information on research design is available in the Nature Research Reporting Summary linked to this article.

## Data availability

For ATR-bound ChRmine in small diameter and large diameter nanodiscs the final models are in the PDB under 7SFK and 7SFJ, the final maps are in the Electron Microscopy Data Bank (EMDB) under EMD-25091 and EMD-25079. For apo-ChRmine, the final model is in the PDB under 7SHS, the final map is in the Electron Microscopy Data Bank (EMDB) under EMD-25135. Original micrograph movies and final particle stack is in the Electron Microscopy Public Image Archive (EMPIAR). Source data are provided with this paper.

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

## Acknowledgements

We thank J. Remis, D. Toso, and P. Tobias for microscope and computational support at the Cal-Cryo facility. We thank members of the Brohawn and Adesnik laboratories for discussions and feedback on the manuscript. H.A. and S.G.B. are New York Stem Cell Foundation-Robertson Neuroscience Investigators. This work was funded by an NSF Graduate Research Fellowship and a UC Berkeley Chancellor's Fellowship (K.T); NIH grant UF1NS107574 (H.A.) and the New York Stem Cell Foundation (H.A. and S.G.B.); and NIGMS grant GM123496, a McKnight Foundation Scholar Award, a Sloan Research Fellowship, a Winkler Family Scholar Award (to S.G.B.).

## Author contributions

K.T. designed and performed all biochemistry and cryo-EM experiments. K.T. performed all cryo-EM data processing. K.T. and S.G.B. modeled and refined the structure. S.S. designed, performed, and analyzed all electrophysiology experiments. H.A. and S.G.B. conceived of and supervised the project. S.G.B. and K.T. wrote the manuscript with input from all authors.

## Competing interests

The authors declare no competing interests.
