## [Peer Review File · Nature Communications]

Cryo-EM structures of the channelrhodopsin ChRmine in lipid nanodiscsREVIEWER COMMENTS

Reviewer #1 (Remarks to the Author):

The manuscript by Tucker et al. entitled “Cryo-EM structures of the channelrhodopsin ChRmine in lipid nanodiscs” reports high-resolution structures of a representative member of the family of “bacteriorhodopsin-like channelrhodopsins”, light-gated cation channels from cryptophytes that likely evolved independently of better known channelrhodopsins from green algae.

This work provides the necessary basis for rational engineering of these channels to meet the needs of optogenetic research and therapy. Since its inception, optogenetics has become a very powerful technology with many diverse applications, so further development of optogenetic tools is expected to be very beneficial.

This manuscript would also be interesting for a more general readership beyond the users of optogenetics, as it expands our mechanistic understanding of light-gated cation channel conductance. Additionally, I would like to praise a very concise, clear writing style of this manuscript. I recommend it for publication in Nature Communications after a minor revision to address the issues listed below:

1) Most importantly, please correct the information on the organismal source of ChRmine in the PDB entries 7SFJ, 7SFK and 7SHS. It is true that ChRmine mRNA is included in the transcriptome of the ciliate *Tiarina fusus* by the Marine Microbial Eukaryote Transcriptome Sequencing Project (MMETSP) and referred to as such in Ref. [17], the authors of which mined the MMETSP database. But this sequence is identical to a BCCR sequence derived from the cryptophyte *Rhodomonas lens* [PMID: 32317325] that was used to feed *Tiarina* prior to RNA extraction by MMETSP. There is no doubt that the real organismal source of ChRmine is *R. lens*.

2) Page 2 (Introduction) “...termed bacteriorhodopsin-like cation channelrhodopsins (BCCRs) because their sequence more closely resembled archaeal pump rhodopsins”.

The name “BCCRs” have been suggested for this protein family not only because their sequences are homologous to those of haloarchaeal proton pumps, but also because they exhibit a closer functional resemblance to the pumps than “classical” ChRs. In particular, Ref. [16] shows that in another BCCR, GtCCR2, cation-channel gating is tightly coupled to intramolecular proton transfers involving the same residues that define the vectorial proton path in bacteriorhodopsin. The retinylidene Schiff base chromophore rapidly deprotonates to the Asp-85 homolog, opening of the cytoplasmic channel requires

the Asp-96 homolog to be unprotonated, and channel closing temporally correlates with reprotonation of the SB.

3) Page 2 (Introduction) "ChRmine is a BCCR <...> distinct among optogenetic tools for its combination of high light sensitivity, cation over proton selectivity, large photocurrents..." (Introduction), and later, "ChRmine's exceptionally large photocurrents" (Results & Discussion).

When ChRmine is probed side by side with other BCCRs in HEK293 cells, its performance is rather mediocre [PMID: 32317325], which means that its large currents in neurons are owing to better expression and/or membrane targeting rather than to its larger unitary conductance. Also, if large photocurrents of ChRmine in neurons resulted from its structural differences with chlorophyte CCRs, all (or many) other, closely homologous BCCRs would generate larger currents than chlorophyte CCRs, which is clearly not the case. So far, ~30 BCCRs have been tested by heterologous expression and patch clamp recording, and most of them generate very small photocurrents.

4) Page 2 (Introduction) "We determined cryo-EM structures of full-length ChRmine in lipid nanodiscs..."

Indeed, the name "ChRmine" has been assigned to a truncated version of this protein (residues 1-309), used as an optogenetic tool. However, the actual algal transcript encodes 456 residues, so referring to the 1-309 version as a "full-length" is confusing. I recommend replace this statement with the construct length (the number of residues).

5) Pages 4-5 (Results & Discussion) "In ChRmine, the pocket around the β -ionone ring is distinctly electronegative due to contributions from S149, E154, and Y113 residues <...>. In the relatively blue-shifted C1C2, however, corresponding residues W201, A206, and Y160 are nonpolar..."

The 3d of these positions (113/160) is occupied by tyrosine in both ChRmine and C1C2. The Authors seem to state that Tyr in this position contributes to the electronegativity of the pocket in ChRmine, but does not in C1C2. This requires an explanation.

5) Page 5 (Results & Discussion) "...extracellular constriction could have opposing effects and increase ChRmine closing kinetics".

Style: Please replace "increase" with "accelerate", or replace "increase ChRmine closing kinetics" with "increase ChRmine closing rate".

6) Page 6 (Results & Discussion) "...the three conduction paths per functional ChRmine channel compared to two in CCRs could contribute to larger photocurrents."

The Authors provide compelling evidence that ChRmine's cation conductance pathway is located within each individual monomer rather than at the trimer interface. I don't understand why they think that grouping monomers into trimers (or dimers, for that matter) would increase the conductance? If the Authors mean some kind of cooperativity between the conductive monomers, they should provide experimental evidence for it, such as comparison of the light dependence of monomers and trimers. If not, I recommend to delete this statement.

7) Page 23 (Figure S5 legend) "Central pore lipids are occluded within the ChRmine trimer"

Style: Is it the passage between the monomers that is occluded by the lipids, not the lipids themselves?

Reviewer #2 (Remarks to the Author):

Overall, this is a very well-written manuscript. The authors really thought their arguments through. Especially when it comes to their structural analysis in comparison to other rhodopsins. That said, the mutation analysis is somewhat underdeveloped and could be improved as indicated in some of the comments below.

1. The explanation for why the mutations were made and what they were expecting to see is lacking. This should be described more clearly. A more complete comparison of the relative impact of these mutations on ChRmine and homologous channelrhodopsins would strengthen the paper.

2. Could the authors better describe the methodology behind the light sensitivity data, including analysis?

3. The Y116W variant only has 2 data points. At the very minimum there should be 3. Aside from that, the photocurrents appear far too small to measure kinetic rates accurately. Therefore it is unclear if any conclusions can be made about the kinetic rates for this variant.

4. The authors also do not report protein expression levels, so they cannot definitively say whether the reduced photocurrent amplitudes were due to reduced expression or reduced conductance. They would need western blots to rule out protein expression issues.

5. Can the authors state whether the CHO cells were incubated with retinal or not?

Reviewer #3 (Remarks to the Author):

The manuscript describes a 3D-structure of the Channelrhodopsin ChRmine, which is the first structure of a CHR in lipids (nanodiscs) and not in detergent. ChRmine, also named RICCR according to its native source organism *Rhodospseudomonas lens*) is the first Cryptophyte ChR (CCR) and attractive due to its conductance that is around 5 times larger than that of most of Chlorophyte ChRs. This class of proteins recently gained even more interest because the recently discovered K⁺-selective KCRs belong to the same CCR family which is more related to the light-driven pumps than other ChRs. The manuscript is clearly written and well understandable.

The manuscript has been published online in competition with another online report about the same protein filed by Kishi 2021, which is my explanation why the analysis of the structure is far less complete as it could be.

Moreover, considering that already a number of ChR-structures have been published (2x C1C2, 2x ChR2, Chrimson, 2x ACR, IC++) the novelty of the manuscript is limited especially owing to the fact that one structure is of a Cryptophyte ACR with large conductance as well.

Comments

1. The structure is sold far below its value because a number of interpretations are wrong or at least incorrect. The authors do not provide any explanation of the unusual properties and no relation with the structure except saying that the pore is wider than that of C1C2. The pores of C1C2 and Chrimson are also wider compared to ChR2 but the conductance is not larger but the selectivity is quite different. A presentation of the ChR2 and Chrimson pores would be interesting. To determine the origin of the enlarged conductance one would need the open pore structure which is not available for any ChR. I agree that the electronegativity of the pore must have an influence on the cation affinity but not necessarily on the conductance; the cations might even stick to the negative charges. Mutations along the pore would support or disprove it but would increase the value of the manuscript substantially.

2. The authors suggest that the flexibility of the binding pocket would explain the slow recovery kinetics and propose to limit the flexibility by voluminous residues - as M201 in Chrimson - to accelerate the closing kinetics. This sounds reasonable but I am skeptical because in the past transfer of specific residues from one to another ChR resulted in many cases in very different properties especially if the ChRs are from different families. The only mutation that accelerated the photocycle was T119A which is a homologue of S169A in Chrimson (and should be mentioned).

3. The authors mention on many occasions that the absorption is red shifted. What does it mean, compared to what ? This was a selling argument in previous publication but with a maximum of around 520 nm (Sineshchekov 2020; Vierock 2021), it is not more red shifted than other ChRs like for example VChR1, ReaChR etc.. The authors should be aware about this previous misinterpretation because they use 510 nm for ChRime activation. They should turn this finding into a more interesting question which means: why does it absorb maximally at 525 nm although the active site is similar to Chrimson (λ_{max} =590nm) with only a single negative counter ion.

The authors could easily provide an absorption spectrum because they have purified protein at hand? The statement that the polarity of the retinal binding pocket might change isomerization should be deleted or needs support. Where does this suggestion come from?

The paragraph is pure speculation and should be substantiated by mutations and owing to the fact that the main absorption is close to 500 nm where most rhodopsins absorb, quite obsolete.

4. The authors mention at many locations in the manuscript that the pore is filled with lipid and is unlikely to conduct ions. This is certainly true but they should not name it "pore" if it is not a pore. This results obviously from the parallel manuscript where the authors see in detergent a water filled pore but the function as such is not shown. I am quite convinced that Tucker et al. interpreted their finding correctly. But, this discussion is not new either because a pore in the center of a dimer or trimer has been discussed for ChR2 by Mueller 2015 and HR with lipids inside that differ from the rest of the membrane lipids by Kolbe et al. 2000.

5. The differences between the functional ChRmine and the apoprotein are surprisingly small. The opening of the cavity is not necessarily expected, why not a collapse? What is IN the binding pocket, is it filled with water?

In summary, the authors do present an interesting structure but do not make much out of it. It is not clear what the major discovery is unless the high ion conductance which should be mechanistically explained or at least interpreted on the basis of a few mutations which should be too much work.

Finally I admire that the pdb-file has been published with the online preprint.

Response to reviewer comments

We thank the reviewers for their thoughtful comments and constructive feedback on our manuscript. Please find below our point-by-point response. In the revised manuscript, we have incorporated requested changes to the text, updated Fig. 3 with comparisons to Chrimson and ChR2, and updated Fig. 4 with data from four new mutant channels (including recently reported hsChRmine and rsChRmine (Kishi et al. Cell 2022)). We show T119A ChRmine shows accelerated closing that is indistinguishable from hsChRmine, but does not suffer from a reduction in photocurrents, suggesting it could serve as a superior optogenetic tool. We hope you will agree the paper is substantially improved.

Reviewer #1 (Remarks to the Author):

The manuscript by Tucker et al. entitled “Cryo-EM structures of the channelrhodopsin ChRmine in lipid nanodiscs” reports high-resolution structures of a representative member of the family of “bacteriorhodopsin-like channelrhodopsins”, light-gated cation channels from cryptophytes that likely evolved independently of better known channelrhodopsins from green algae.

This work provides the necessary basis for rational engineering of these channels to meet the needs of optogenetic research and therapy. Since its inception, optogenetics has become a very powerful technology with many diverse applications, so further development of optogenetic tools is expected to be very beneficial.

This manuscript would also be interesting for a more general readership beyond the users of optogenetics, as it expands our mechanistic understanding of light-gated cation channel conductance. Additionally, I would like to praise a very concise, clear writing style of this manuscript. I recommend it for publication in Nature Communications after a minor revision to address the issues listed below:

1) Most importantly, please correct the information on the organismal source of ChRmine in the PDB entries 7SFJ, 7SFK and 7SHS. It is true that ChRmine mRNA is included in the transcriptome of the ciliate *Tiarina fusus* by the Marine Microbial Eukaryote Transcriptome Sequencing Project (MMETSP) and referred to as such in Ref. [17], the authors of which mined the MMETSP database. But this sequence is identical to a BCCR sequence derived from the cryptophyte *Rhodomonas lens* [PMID: 32317325] that was used to feed *Tiarina* prior to RNA extraction by MMETSP. There is no doubt that the real organismal source of ChRmine is *R. lens*.

Thank you for pointing out our oversight. We have corrected the PDB entries and included the correct source organism in the methods.

2) Page 2 (Introduction) “...termed bacteriorhodopsin-like cation channelrhodopsins (BCCRs) because their sequence more closely resembled archaeal pump rhodopsins”.

The name “BCCRs” have been suggested for this protein family not only because their sequences are homologous to those of haloarchaeal proton pumps, but also because they exhibit a closer functional resemblance to the pumps than “classical” ChRs. In particular, Ref. [16] shows that in another BCCR, GtCCR2, cation-channel gating is tightly coupled to intramolecular proton transfers involving the same residues that define the vectorial proton path in bacteriorhodopsin. The retinylidene Schiff base chromophore rapidly deprotonates to the Asp-85 homolog, opening of the cytoplasmic channel requires the Asp-96 homolog to be unprotonated, and channel closing temporally correlates with reprotonation of the SB.

The introduction has been updated to indicate BCCRs more closely resemble pump rhodopsins than CCRs by sequence and mechanism as shown in Ref. 16.

3) Page 2 (Introduction) “ChRmine is a BCCR <...> distinct among optogenetic tools for its combination of high light sensitivity, cation over proton selectivity, large photocurrents...” (Introduction), and later, “ChRmine’s exceptionally large photocurrents” (Results & Discussion).

When ChRmine is probed side by side with other BCCRs in HEK293 cells, its performance is rather mediocre [PMID: 32317325], which means that its large currents in neurons are owing to better expression and/or

membrane targeting rather than to its larger unitary conductance. Also, if large photocurrents of ChRmine in neurons resulted from its structural differences with chlorophyte CCRs, all (or many) other, closely homologous BCCRs would generate larger currents than chlorophyte CCRs, which is clearly not the case. So far, ~30 BCCRs have been tested by heterologous expression and patch clamp recording, and most of them generate very small photocurrents.

We have included text in the revision stating that the relative contribution of conductance and expression/membrane targeting differences to large photocurrents of ChRmine relative to other channelrhodopsins that has been reported (e.g. Sridharan et al. *Neuron* 2022) is unknown. We also include the point raised above and reference Sineshchekov et al. *Mbio* 2020.

4) Page 2 (Introduction) “We determined cryo-EM structures of full-length ChRmine in lipid nanodiscs...”

Indeed, the name “ChRmine” has been assigned to a truncated version of this protein (residues 1-309), used as an optogenetic tool. However, the actual algal transcript encodes 456 residues, so referring to the 1-309 version as a “full-length” is confusing. I recommend replace this statement with the construct length (the number of residues).

Thank you for this suggestion. We have updated the text to make this clear.

5) Pages 4-5 (Results & Discussion) “In ChRmine, the pocket around the β -ionone ring is distinctly electronegative due to contributions from S149, E154, and Y113 residues <...>. In the relatively blue-shifted C1C2, however, corresponding residues W201, A206, and Y160 are nonpolar...”

The 3d of these positions (113/160) is occupied by tyrosine in both ChRmine and C1C2. The Authors seem to state that Tyr in this position contributes to the electronegativity of the pocket in ChRmine, but does not in C1C2. This requires an explanation.

Thank you for pointing out this poor wording. We have edited the text to state: “In the relatively blue-shifted C1C2, however, two of three corresponding residues (W201 and A206) are nonpolar or positioned further away from the retinal, resulting in a less electronegative retinal binding pocket.”

6) Page 5 (Results & Discussion) “...extracellular constriction could have opposing effects and increase ChRmine closing kinetics”.

Style: Please replace “increase” with “accelerate”, or replace “increase ChRmine closing kinetics” with “increase ChRmine closing rate”.

Corrected.

7) Page 6 (Results & Discussion) “...the three conduction paths per functional ChRmine channel compared to two in CCRs could contribute to larger photocurrents.”

The Authors provide compelling evidence that ChRmine’s cation conductance pathway is located within each individual monomer rather than at the trimer interface. I don’t understand why they think that grouping monomers into trimers (or dimers, for that matter) would increase the conductance? If the Authors mean some kind of cooperativity between the conductive monomers, they should provide experimental evidence for it, such as comparison of the light dependence of monomers and trimers. If not, I recommend to delete this statement.

We have deleted this statement.

8) Page 23 (Figure S5 legend) “Central pore lipids are occluded within the ChRmine trimer”

Style: Is it the passage between the monomers that is occluded by the lipids, not the lipids themselves?

Both are true. The pore lipids block ion conduction through the passage. The lipids are also occluded from bulk

lipids by intracellular ChRmine extensions that form a protein “fence” between the central pore and the surrounding membrane. The legend has been edited to make this clear.

Reviewer #2 (Remarks to the Author):

Overall, this is a very well-written manuscript. The authors really thought their arguments through. Especially when it comes to their structural analysis in comparison to other rhodopsins. That said, the mutation analysis is somewhat underdeveloped and could be improved as indicated in some of the comments below.

1. The explanation for why the mutations were made and what they were expecting to see is lacking. This should be described more clearly. A more complete comparison of the relative impact of these mutations on ChRmine and homologous channelrhodopsins would strengthen the paper.

Thank you for this suggestion. We have included a more thorough explanation for why mutants were made, expected results, and interpretation. We have also included analysis of four additional mutants in the revised paper (Fig. 4).

2. Could the authors better describe the methodology behind the light sensitivity data, including analysis?

Light sensitivity is reported as ED50 from a five-parameter logistic regression to plots of photocurrent versus measured light intensity recorded from successive sweeps with varying LED power. We had reported sensitivity as the percentage of maximum LED power, but replaced this with the corresponding measured light intensity in mW/mm². Additional text has been added to the methods section.

3. The Y116W variant only has 2 data points. At the very minimum there should be 3. Aside from that, the photocurrents appear far too small to measure kinetic rates accurately. Therefore it is unclear if any conclusions can be made about the kinetic rates for this variant.

All ChRmine constructs now have n≥3 cells. Y116W indeed show low currents, but a zoomed view of a trace shows kinetics are still estimated with reasonable accuracy.

4. The authors also do not report protein expression levels, so they cannot definitively say whether the reduced photocurrent amplitudes were due to reduced expression or reduced conductance. They would need western blots to rule out protein expression issues.

We agree and now state this explicitly in the text.

5. Can the authors state whether the CHO cells were incubated with retinal or not?

Yes, cells were incubated with 5 μM all-trans-retinal. This has been added to the methods.

Reviewer #3 (Remarks to the Author):

The manuscript describes a 3D-structure of the Channelrhodopsin ChRmine, which is the first structure of a CHR in lipids (nanodiscs) and not in detergent. ChRmine, also named RICCR according to its native source organism *Rhodospseudomonas lens* is the first Cryptophyte ChR (CCR) and attractive due to its conductance that is around 5 times larger than that of most of Chlorophyte ChRs. This class of proteins recently gained even more interest because the recently discovered K⁺-selective KCRs belong to the same CCR family which is more related to the light-driven pumps than other ChRs. The manuscript is clearly written and well understandable.

The manuscript has been published online in competition with another online report about the same protein filed by Kishi 2021, which is my explanation why the analysis of the structure is far less complete as it could be. Moreover, considering that already a number of ChR-structures have been published (2x C1C2, 2x ChR2, Chrimson, 2x ACR, IC++) the novelty of the manuscript is limited especially owing to the fact that one structure is of a Cryptophyte ACR with large conductance as well.

Comments

1. The structure is sold far below its value because a number of interpretations are wrong or at least incorrect. The authors do not provide any explanation of the unusual properties and no relation with the structure except saying that the pore is wider than that of C1C2. The pores of C1C2 and Chrimson are also wider compared to ChR2 but the conductance is not larger but the selectivity is quite different. A presentation of the ChR2 and Chrimson pores would be interesting. To determine the origin of the enlarged conductance one would need the open pore structure which is not available for any ChR. I agree that the electronegativity of the pore must have an influence on the cation affinity but not necessarily on the conductance; the cations might even stick to the negative charges. Mutations along the pore would support or disprove it but would increase the value of the manuscript substantially.

We agree an open pore structure is an essential next step to fully understand determinants of channel properties. A revised Fig. 3 now shows a comparison between ChRmine, C1C2, Chrimson, and ChR2. One interpretation of reduced photocurrents in the H33R mutation (included in revised Fig. 4) is that constriction of the pore by the larger basic residue reduces conductance. Still, we point out that we cannot exclude other interpretations (e.g. reduced functional expression) and this hypothesis needs to be further tested in future work.

2. The authors suggest that the flexibility of the binding pocket would explain the slow recovery kinetics and propose to limit the flexibility by voluminous residues - as M201 in Chrimson - to accelerate the closing kinetics. This sounds reasonable but I am skeptical because in the past transfer of specific residues from one to another ChR resulted in many cases in very different properties especially if the ChRs are from different families. The only mutation that accelerated the photocycle was T119A which is a homologue of S169A in Chrimson (and should be mentioned).

Thank you for pointing out the reference to ChRmine S169A was omitted in the section describing to ChRmine T119A. This has been corrected.

We agree transferring mutations between channelrhodopsins is not straightforward. In ChRmine, additional mutagenesis supports the idea that a tighter retinal binding pocket results in faster closing like in Chrimson. Mutants I146M/G174S, G174T, and F178S were predicted to result in tighter retinal binding pockets and all show significantly faster closing rate (ChRmine I146M is analogous to ChrimsonM201) (Fig. 4).

3. The authors mention on many occasions that the absorption is red shifted. What does it mean, compared to what? This was a selling argument in previous publication but with a maximum of around 520 nm (Sineshchekov 2020; Vierock 2021), it is not more red shifted than other ChRs like for example VChR1, ReaChR etc.. The authors should be aware about this previous misinterpretation because they use 510 nm for ChRmine activation. They should turn this finding into a more interesting question which means: why does it absorb maximally at 525 nm although the active site is similar to Chrimson ($\lambda_{max} = 590\text{nm}$) with only a single negative counter ion.

The authors could easily provide an absorption spectrum because they have purified protein at hand? The statement that the polarity of the retinal binding pocket might change isomerization should be deleted or needs support. Where does this suggestion come from?

The paragraph is pure speculation and should be substantiated by mutations and owing to the fact that the main absorption is close to 500 nm where most rhodopsins absorb, quite obsolete.

We mean ChRmine is red shifted relative to prototypical ChR2 and are careful not to imply it is more red-shifted than ReaChR or Chrimson (an absorbance spectrum is now published in Kishi et al. *Cell* 2022). We indicate polarity of the retinal binding pocket influencing spectral properties is speculative and reference previous literature supporting the idea.

4. The authors mention at many locations in the manuscript that the pore is filled with lipid and is unlikely to conduct ions. This is certainly true but they should not name it "pore" if it is not a pore. This results obviously from the parallel manuscript where the authors see in detergent a water filled pore but the function as such is not shown. I am quite convinced that Tucker et al. interpreted their finding correctly. But, this discussion not

new either because a pore in the center of a dimer or trimer has been discussed for ChR2 by Mueller 2015 and HR with lipids inside that differ from the rest of the membrane lipids by Kolbe et al. 2000.

We agree, reference classic work on HR (Kolbe et al. *Nature* 2000) as an analogous lipid-filled central pore, and point out differences between the central pore HR and ChRmine. We still prefer to call it a pore even though half is filled with lipid. We think it is important to be clear in our interpretation of these data because the Kishi et al. preprint argued for conduction through the central pore based on their detergent solubilized structure.

5. The differences between the functional ChRmine and the apoprotein are surprisingly small. The opening of the cavity is not necessarily expected, why not a collapse? What is IN the binding pocket, is it filled with water?

We agree the differences are small and speculate the retinal binding pocket is water-filled in the apo structure, but cannot be certain because waters are not observed given the lower resolution of these data.

In summary, the authors do present an interesting structure but do not make much out of it. It is not clear what the major discovery is unless the high ion conductance which should be mechanistically explained or at least interpreted on the bases of a few mutations which should be too much work.

Finally I admire that the pdb-file has been published with the online preprint.

Thank you, we think this is important and hope becomes more common.

REVIEWERS' COMMENTS

Reviewer #1 (Remarks to the Author):

The Authors have taken into account all my comments and the comments of other reviewers, and provided answers to all our questions in the revised version. I recommend the revised version for publication.

Reviewer #2 (Remarks to the Author):

The authors have satisfactorily addressed the questions from this reviewer.